# Prognostic Factors for Overall Survival in Advanced Intrahepatic Cholangiocarcinoma Treated with Yttrium-90 Radioembolization

**DOI:** 10.3390/jcm9010056

**Published:** 2019-12-25

**Authors:** Michael Köhler, Fabian Harders, Fabian Lohöfer, Philipp M. Paprottka, Benedikt M. Schaarschmidt, Jens Theysohn, Ken Herrmann, Walter Heindel, Hartmut H. Schmidt, Andreas Pascher, Lars Stegger, Kambiz Rahbar, Moritz Wildgruber

**Affiliations:** 1Department of Clinical Radiology, Universitätsklinikum Münster, D-48149 Münster, Germany; michael.koehler@ukmuenster.de (M.K.); Fabian.harders@t-online.de (F.H.); heindel@uni-muenster.de (W.H.); 2Network Partner Site Westdeutsches Tumorzentrum, D-45147 Essen, Germany; Benedikt.Schaarschmidt@uk-essen.de (B.M.S.); jens.theysohn@uk-essen.de (J.T.); Ken.Herrmann@uk-essen.de (K.H.); hepar@ukmuenster.de (H.H.S.); andreas.pascher@ukmuenster.de (A.P.); lars.stegger@ukmuenster.de (L.S.); kambiz.rahbar@ukmuenster.de (K.R.); 3Division of Interventional Radiology, Klinikum rechts der Isar der Technischen Universität München, D-81675 München, Germany; fabian.lohoefer@tum.de (F.L.); philipp.paprottka@tum.de (P.M.P.); 4Department for Diagnostic and Interventional Radiology, Universitätsklinikum Essen, D-41547 Essen, Germany; 5Department for Nuclear Medicine, Universitätsklinikum Essen, D-41547 Essen, Germany; 6Department of Gastroenterology and Hepatology, Universitätsklinikum Münster, D-48149 Münster, Germany; 7Department for General, Visceral and Transplantation Surgery, Universitätsklinikum Münster, D-48149 Münster, Germany; 8Department of Nuclear Medicine, Universitätsklinikum Münster, D-48149 Münster, Germany

**Keywords:** intrahepatic cholangiocarcinoma, radioembolization, Y-90 Yttrium

## Abstract

Purpose: To evaluate factors associated with survival following transarterial ^90^Y (yttrium) radioembolization (TARE) in patients with advanced intrahepatic cholangiocarcinoma (ICC). Methods: This retrospective multicenter study analyzed the outcome of three tertiary care cancer centers in patients with advanced ICC following resin microsphere TARE. Patients were included either after failed previous anticancer therapy, including relapse after surgical resection, or for having a minimum of 25% of total liver volume affected by ICC. Patients were stratified and response was assessed by the Response Evaluation Criteria in Solid Tumors (RECIST) criteria at 3 months. Kaplan–Meier analysis was performed to analyze survival followed by cox regression to determine independent prognostic factors for survival. Results: 46 patients were included (19 male, 27 female), median age 62.5 years (range 29–88 years). A total of 65% of patients had undergone previous therapy, while 63% had a tumor volume > 25% of the entire liver volume. Median survival was 9.5 months (95% CI: 6.1–12.9 months). Due to loss in follow-up, *n* = 37 patients were included in the survival analysis. Cox regression revealed the extent of liver disease to one or both liver lobes being associated with survival, irrespective of tumor volume (*p* = 0.041). Patients with previous surgical resection of ICC had significantly decreased survival (3.9 vs. 12.8 months, *p* = 0.002). No case of radiation-induced liver disease was observed. Discussion: Survival after ^90^Y TARE in patients with advanced ICC primarily depends on disease extent. Only limited prognostic factors are associated with a general poor overall survival.

## 1. Introduction

Intrahepatic cholangiocarcinoma represents the second most common primary liver cancer after hepatocellular carcinoma. The reported incidence has increased worldwide over the past few decades [1]. Mortality from ICC is increasing in the US and UK and has tripled in Germany from 1998 to 2008 [2,3]. The five-year survival is below 5%, even for those patients who undergo tumor resection. Surgical resection is the only potentially curative therapy option for patients with ICC. However, only 20% are eligible for resection. The reason for the low resectability rate is disease spread and anatomic location, inadequate hepatic reserve or comorbidities [2]. The median survival for patients with untreated unresectable ICC has been reported to be as low as 3–6 months [4]. Systemic chemotherapy has a low survival benefit for patients with unresectable ICC, as the majority of patients have a chemorefractory course. For systemic chemotherapies like gemcitabine or gemcitabine and cisplatin, the previously reported survival rate is between 8.1 and 11.7 months, respectively [5]. A recent post-hoc analysis of three large clinical trials just recently revealed improved survival rates of 16.7 months after 3 or 6 months of chemotherapy [6]. Thus, the therapeutic options for patients suffering from advanced ICC are associated with poor response rates and severe side effects at the same time. To date, there is no consensus on the optimal treatment regime of unresectable and chemorefractory ICC [7]. The goals of palliative therapy are clearly defined; in detail, these are control of local tumor growth, to relieve tumor induced symptoms, and to improve the quality of life.

There are a number of different local-ablative treatment options for unresectable and chemorefractory ICC, i.e., radiofrequency ablation (RFA), intra-arterial chemotherapy, transarterial chemoembolization (TACE), and radioembolization (TARE). Since the first reports, ^90^Y radioembolization has been shown to be an effective method for both primary liver cancer, especially HCC, as well as secondary liver cancer, with the most existing evidence in metastatic colorectal cancer [8,9]. There is an increasing number of reports on the use of ^90^Y radioembolization for intrahepatic cholangiocarcinoma, but the evidence of resin-based ^90^Y microspheres in the treatment of unresectable and chemorefractory ICC is still limited [3,10,11,12,13,14]. First results suggest that the median overall survival treating ICC with ^90^Y radioembolization is 13.9 months (9.5–18.3 months), thus being at least as high compared to systemic chemotherapy, but potentially reduced side effects [3]. Most reported results confer to therapy-naïve patients when TARE is a first line therapy. In clinical practice, however, this is frequently not the case, and patients have undergone multiple therapies before presenting for ^90^Y-TARE.

We therefore aimed to further evaluate the safety and outcome of ^90^Y radioembolization for treatment of advanced intrahepatic cholangiocarcinoma in a retrospective multicenter cohort.

## 2. Materials and Methods

### 2.1. Study Design

The study was carried out as a retrospective multicenter observational trial in three tertiary care academic medical centers in Germany. The study was approved by the local ethics committee (Ethics committee of the Westfalian Wilhems University Münster, protocol number 2018–638–f-S, approval date 19th November 2018).

### 2.2. Patient Selection

All patients undergoing ^90^Y radioembolization had approval of the gastrointestinal tumor board. The basis for the diagnosis of ICC was the diagnosis criteria of the European Association for the Study of the Liver Disease (EASL), and the diagnosis was confirmed in all patients by either endoscopic or percutaneous biopsy.

Patients enrolled met the following criteria: (1) >18 years old; (2) Eastern Cooperative Oncology Group (ECOG) performance scores were ≤2; (3) definite diagnosis of ICC following the EASL guideline [7]; (4) presenting with an advanced disease stage, either refractory to previous systemic therapy or relapse after surgical resection, or—if therapy-naïve-presenting, with a tumor burden of >25% of total liver volume; and (5) adequate hepatic function as follows: Child-Pugh liver function grade A or B, ALT and AST ≤ 5 × upper limit of normal (ULN) and total bilirubin ≤ 1.5 × ULN and albumin ≥ 29 g/L. Moreover, life expectancy was supposed to be more than 12 weeks. Extrahepatic disease was not considered an exclusion criterion but had to be limited to lymph node metastasis. Additionally, extrahepatic disease had to be considered not to be a life-limiting factor when compared to the primary tumor. Patients with solid organ metastasis and peritoneal carcinomatosis were not considered suitable for radioembolization. Hepatopulmonary shunt fraction had to be less than 20%. Adequate hematologic, clotting, and renal function tests were a precondition.

Patients with a complete obstruction of the main portal vein and patients with secondary malignancy were excluded.

### 2.3. Procedure Details 

^90^Y radioembolization using resin microspheres (Sirtex Medical Europe, Bonn, Germany) was performed according to standard operating procedures. Evaluation for the preparation of radioembolization was done using ^99m^Tc-MAA (Technetium-macro albumin aggregated) after embolization of aberrant vasculature originating from the hepatic circulation. Subsequent to ^99m^Tc-MAA application, patients underwent planar whole body and SPECT/CT (single photon emission computed tomography) scanning of the thoracic and abdominal region (GE Discovery NM630 or Siemens Symbia T2) using low energy collimators for dose calculation, detection of extrahepatic tracer accumulation, and assessment of hepatopulmonary shunting. The maximum pulmonary shunt fraction accepted was 20%. The ^90^Y dose was calculated by physicists and board-certified nuclear medicine physicians in the department of nuclear medicine. The body surface area (BSA) method was used for dose calculation (Activity of SIR-Spheres in GBq = (BSA − 0.2) + (Volume of Tumor/Volume of whole liver)).

Radioembolization using ^90^Y resin microspheres (SIR-Spheres^®^; Sirtex Medical, Sydney, Australia) was performed in an either monolobar (one liver lobe treated only), bilobar selective (treatment of both lobes consecutively with the microcatheter positioned just after the bifurcation of the left and right hepatic artery), bilobar superselective (both lobes treated, however with the microcatheter positioned distally in tumor-feeding vasculature only, thereby sparing parts of non-affected liver tissue) or from a central catheter position (with the microcatheter placed before the bifurcation of the right and left hepatic artery) approach. Actual TARE therapy was performed 16.9 ± 8.2 days following initial angiographic evaluation.

The angiographic therapeutic procedures were solely performed by interventional radiologists who were approved within the quality assurance program by Sirtex Medical in cooperation with nuclear medicine specialists responsible for the microsphere administration. Either a ^90^Y Bremsstrahl PET/CT or PET/MRI scan was performed following the procedures for documentation of the regional distribution of radioactivity within the liver.

Diagnostic imaging for follow-up after therapy was performed with biphasic dynamic contrast-enhanced CT or triphasic contrast-enhanced MR with hepatocellular phase imaging.

## 3. Data Collection

All patient and procedural data were retrospectively acquired from electronic patient records as well as from the Picture Archiving and Communications System (PACS). Tumor responses were based on comparative evaluation of pre- and post-treatment scans and evaluated by the Response Evaluation Criteria in Solid Tumors (RECIST), version 1.1. Analysis was performed using Philips Intellispace portal software, version 11 (Philips, Best, The Netherlands).

### Statistical Analysis

Data are shown as total number and percentage, mean and standard deviation or median and range or 95% confidence interval (CI), as appropriate. Kaplan–Meier analysis was performed to analyze survival, and a subsequent log-rank test was applied to determine prognostic factors that influence survival. For parameters with independent impact on survival, a cox regression was applied to determine the hazard ratio for increased risk of death within 12 months following radioembolization therapy. A *p*-value < 0.05 was considered statistically significant.

## 4. Results

### 4.1. Patient Characteristics

A total of 46 patients were included in this retrospective cohort study (19 male, 27 female). The patients had a median age of 62.5 years (range 29 to 88 years). Further baseline characteristics of patients and tumors are summarized in Table 1. A total of 28 patients (61%) had undergone previous therapy for ICC (Table 2). All patients where previous therapy radioembolization was performed due to a missing response to those previous therapies either had stable or progressive disease with viable tumor being present on cross-sectional imaging. In 3 cases (6.5%), the left lobe only was affected, in 14 (30.5%) cases the right lobe, and in 29 patients (63%) both lobes were affected (diagnosed by computed tomography and confirmed by endoscopic retrograde cholangiopancreatography). A total of 29 patients (63%) had a tumor volume of >25% of total liver volume. Baseline laboratory values are listed in Table 3. Three patients had undergone previous treatment with ^90^Y TARE with initial positive response (SD or PR).

### 4.2. Procedural Characteristics

All radioembolization procedures were performed on an inpatient basis. Application of resin TARE spheres was performed either in a monolobar, bilobar or from a central catheter position. Application was in all cases performed via a coaxial microcatheter in all cases, either in a selective or superselective manner. Median applied dose was 1.74 GBq (Table 4).

### 4.3. Tumor Response and Survival

A total of 8/46 patients (17.4%) were lost to follow-up. Local tumor control at 3 months was achieved in 23 (50%) patients, while 13 patients (28.3%) showed progressive disease (Table 5). Two patients (4.3%) had died at 3 months. The subsequent analyses of potential prognostic factors influencing survival are generally limited due to the decreased sample size, and further analysis of subgroups was not possible with an adequate statistical power.

Median survival was 9.5 months (95% CI: 6.1–12.9 months) with a nonsignificant prolonged survival in female compared to male patients (11.2 vs. 5.1 months, *p* = 0.331, Table 6). Kaplan–Meier analysis revealed that tumor extent (uni- vs. bilobar disease) and previous surgical resection are independent factors associated with survival. If disease was limited to one liver lobe, survival was substantially increased compared to patients suffering from bilobar tumor spread (15.6 vs. 6.4 months, *p* = 0.042, Figure 1A), which was confirmed as an independent predictor in a subsequent cox regression analysis revealing a Hazard Ratio HR of 3.4 for dying within the observation period (*p* = 0.012, Table 7). Additionally, patients who had undergone surgical resection show decreased survival (3.9 vs. 12.8 months, *p* = 0.001, Figure 1B) with an HR of 6.2 for death in the observation period (*p* = 0.001). Similarly, patients with systemic previous systemic therapies showed decreased survival (7.3 vs. 18.8 months), however without reaching statistical significance (*p* = 0.054, Figure 1C). Interestingly, the presence of extrahepatic metastasis did not influence survival, indicating that the extent of liver disease to both lobes predominantly determines the course of disease and prognosis instead of mere tumor volume. Similarly, the presence of cirrhosis, portal vein invasion/thrombosis, extent of hepatopulmonary shunt fraction, and baseline bilirubin and albumin levels had no prognostic value on survival. In the entire multicenter retrospective cohort, no case of radiation induced liver disease was observed. A total of *n* = 3 patients who had undergone ^90^Y TARE twice did not experience a significant decline in liver function tests.

## 5. Discussion

Tumors of the biliary tree are associated with a poor prognosis [5,7,15]. While surgical techniques have expanded in recent years, allowing for curative approaches even for advanced Klatskin tumors applying procedures such as associating liver partition and portal vein ligation (ALPPS) and portal vein embolization (PVE) with subsequent extended liver resection, intrahepatic cholangiocarcinoma can rarely be resected in a curative manner [16,17]. Moreover, biliary tract cancer shows poor response rates to systemic chemotherapy, with overall survival previously reported for gemcitabine monotherapy of 8.1 months and gemcitabine plus cisplatin of 11.7 months [5]; 13.8 months for intrahepatic cholangiocarcinoma [18] and 16.2 months for hepatocholangiocarcinoma [19]. A recent post-hoc analysis of three large clinical trials on advanced ICC showed survival rates of 16.7 months following 3 to 6 months of chemotherapy, which likely reflects improved patient selection and improvements in chemotherapy regimens [6]. Overall survival for traditional local ablative therapies such as TACE is similarly poor, with an overall survival after of 10–15 months [20]. For radiofrequency ablation, survival rates of 23.6 months have been reported, with the caveat that radiofrequency ablation is limited to tumors with a small to moderate size, 1.8 cm in the cohort reported by Takahashi et al. [21].

Radioembolization has been proposed as a potential alternative for irresectable ICC, particularly because TARE especially in hypovascular liver tumors is considered more suitable than other transarterial embolization therapies, as the chemotherapeutic agents might not adequately reach the target in poorly perfused lesions [3,22]. Several studies have reported a potential benefit for TARE in ICC, but most confer to patients with limited disease or naïve to other anticancer treatments [3,12]; in some, TARE was able to even achieve a disease reduction allowing for subsequent surgical resection [14]. A recent pooled analysis of ^90^Y TARE in unresectable ICC summarizing 12 cohort studies revealed a median survival of 14.3 months, similarly indicating that most TARE studies included patients with a limited disease extent [23]. A small study of 23 patients investigated differences in response to TARE in therapy-naïve ICC patients compared to patients with previous therapy and showed that survival is substantially increased in therapy-naïve patients undergoing Y-90 TARE [22].

Therefore, the primary aim of the presenting study was to investigate TARE in the treatment of more advanced stages of disease. Patients either refractory to previous anticancer therapy or, if therapy-naïve, with at least a tumor burden of 25% of the entire liver volume were included. In the presented patient cohort with an advanced disease stage, median survival was 9.5 months. At 3 months post TARE, 50% patients achieved local tumor control with partial remission or stable disease and 28% patients suffering from progressive disease. These results confirm the previous findings of Mosconi et al., who described survival rates of 16 months in 19 patients with advanced disease stage having undergone previous anticancer therapy, compared to 52 months in *n* = 4 therapy-naïve ICC patients [22]. In the analysis of prognostic factors for decreased survival, patients having undergone previous surgical resection for ICC had especially low survival rates of four months only after TARE. This does not necessarily mean that TARE is not indicated in those patients, but the limited benefit needs to be balanced carefully with associated risks. Additionally, previous surgical therapy can rather be seen as a surrogate marker for relapse or progressive disease, rather than a real prognostic factor for survival. Interestingly, a predictive model based on 405 patients with biliary tract cancer revealed improved survival of patients who had undergone surgical resection of the primary tumor [24]. However, it has to be noted that this cohort included patients with various biliary tract cancers, including extrahepatic manifestation, in whom an R0 resection can be achieved much more frequently. Moreover, this patient cohort potentially suffered from a less aggressive form of disease, whereas our cohort was focused on advanced stages of intrahepatic cholangiocarcinoma only.

Previous systemic therapies just missed statistical significance as a prognostic factor in our study (*p* = 0.054), which is attributed to the limited sample size and, therefore, a decreased statistical power. However, there is a clear trend that previous systemic therapy is associated with decreased survival (7.3 vs. 13.1 months in our study), which has been confirmed by other studies [22]. Similarly as above, however, previous systemic therapy can be rather considered as a surrogate marker for progressive disease. Moreover, a bilobar tumor extent was associated with decreased survival rates compared to unilobar disease only, while mere tumor volume interestingly did not show to be a prognostic factor for survival after TARE in advanced ICC. Similar, the presence of extrahepatic disease had no impact on survival, indicating that the intrahepatic tumor spread remains the major determining factor for survival. This also suggests that TARE may be indicated in patients with limited metastasis and with liver-dominant disease, which has to evaluated in further studies. Serum levels of liver function such as albumin and bilirubin as well as the tumor markers did not correlate with survival in our study. Of note, only a minority of patients had impaired liver function in our cohort reflected by normal bilirubin levels in most patients. Interestingly, Edeline et al. found carcinoembryogenic antigen (CEA) to be a prognostic factor in advanced ICC undergoing combined radioembolization and chemotherapy [25].

To date, there is no evidence that the choice of glass versus resin microspheres has a significant impact on tumor response or survival [3,10,11,22]. However, Levillain et al. showed that the efficacy of TARE in unresectable ICC significantly depends on the delivered tumor radiation dose. Mean tumoral radiation dose was higher with the partition model applied compared to the body-surface area model, suggesting that a personalized radiation activity regime should be performed for each patient [13].

While in the present study, ^90^Y TARE was applied as a salvage therapy option for ICC, when other anticancer treatments including surgery had failed or disease relapse occurred, ^90^Y TARE might be combined with other anticancer treatments in a multimodality setting earlier in the course of the disease. A recent phase Ib trial of combined gemcitabine chemotherapy and ^90^Y TARE has proven an acceptable safety profile for gemcitabine doses up to 600 mg/m^2^, with all patients experiencing grade 1 toxicity, 37.5% grade 2, and 12.5% grade 3 toxicities [26]. Survival data of the study are still pending. These multimodal therapy regimes, including ^90^Y TARE, have the potential to improve survival and reduce side effects compared to sole chemotherapeutic approaches using high dose regimens. Furthermore, the advent of new chemotherapy regimens with improved survival rates [6], immunotherapies, as well as improved patient selection, promise novel multimodal treatment options when combined with ^90^Y TARE for the treatment of ICC [27,28,29]. The first prospective trial combining radioembolization and chemotherapy (again with cisplatin and gemcitabine), which was recently published, reported a median overall survival of 22 months following combined treatment [25]. The improved survival rates in this study compared to our cohort is attributed again to the much-advanced tumor stages and similarly increased morbidity in our cohort, having undergone already multiple therapies, while the study cohort of Edeline et al. were all therapy-naïve when included in the protocol. A remarkable result of the study by Edeline et al. was that 9/41 patients (previously not suited for surgical resection) were able to undergo curative surgery following downstaging with combined radioembolization and chemotherapy.

This study is limited by its retrospective character, including a significant amount of loss to follow-up at 12 months, and additionally by the limited sample size of 47 patients. To overcome this problem, data were pooled from three collaborating tertiary care cancer centers to increase the size of the analyzed cohort. Nonetheless, the collected patient cohort was still heterogenic, especially with regard to previous therapies, limiting the ability to perform adequate subgroup analyses according to specific pre-treatments. The presented sample size limits the power of the analysis of potential prognostic factors, and interpretation of the retrieved results has to be performed with caution. Previous surgical as well as systemic therapies in patients with advanced ICC, moreover, have to be rather interpreted as surrogate markers of progressive disease, which was present in the patients with radioembolization being performed following previous therapies. Further studies with higher patient numbers and subsequently increased power are expected to reveal further prognostic factors, such as previous systemic therapies, which missed statistical significance in our cohort and which were identified as relevant in other cohorts. Additionally, is has to be emphasized that *n* = 3 patients received a repeated radioembolization within the observation period, which induces a certain bias.

The poor overall survival in the presented study cohort raises the question of whether TARE in patients with progressive disease or disease relapse after previous therapy has substantial benefit compared to best supportive care alone. This question, however, can only be answered through prospective randomized trials and, as already discussed, embedding radioembolization in a multimodal therapy regimen in patients with advanced ICC has to be further explored in subsequent research.

## 6. Conclusions

The present study shows that in an advanced disease stage of ICC, ^90^Y TARE can be applied in a salvage approach, however with poor overall survival, which depends primarily on disease extent.

## Figures and Tables

**Figure 1 jcm-09-00056-f001:**
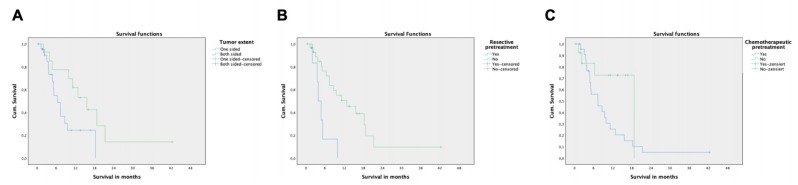
Kaplan–Meier Analysis of survival after ^90^Y TARE. (**A**) Increased cumulative survival of patients with unilobar (*n* = 17) versus bilobar (*n* = 29) disease (*p* = 0.042). (**B**) Increased survival of patients without previous resective surgery (*n* = 28) compared to patients who have undergone previous resection (*n* = 9, *p* = 0.002); a total of *n* = 37 patients with sufficient follow-up data were included in the analysis. (**C**) Previous systemic therapy missed statistical significance as a prognostic factor in this cohort of patients.

**Table 1 jcm-09-00056-t001:** Baseline patient characteristics.

Variables	No. of Patients (Percentage %)
Sex	
Male	19 (41.3)
Female	27 (58.7)
Liver cirrhosis	
Yes	5 (10.9)
No	41 (89.1)
Child-Pugh-Score	
A	2 (4.4)
B	3 (6.5)
C	0 (0.0)
Tumor burden	
<25%	17 (37.0)
25–50%	26 (56.5)
>50%	3 (6.5)
Affected Lobes	
Right liver lobe	14 (30.5)
Left liver lobe	3 (6.5)
Bilobar	29 (63.0)
Portal Vein Thrombosis	
Yes	14 (30.4)
No	32 (69.6)
Extrahepatic Tumor	
Yes	14 (30.4)
No	32 (69.6)
Bilirubin	
Normal	43 (93.5)
Increased	3 (6.5)

**Table 2 jcm-09-00056-t002:** Locoregional and systemic therapies.

Variables	No. of Patients (Percentage %)
Previous therapy	
Yes	30 (65.2)
No	16 (34.8)
Systemic Chemotherapy	28 (93.3)
Immunotherapy	1 (3.3)
Radiation	4 (13.3)
Resection	9 (30.0)
TACE	1 (3.3)
Follow up on therapies	
Remission	0 (0.0)
Stable Disease	2 (6.7)
Progression	28 (3.3)

TACE = transarterial chemoembolization.

**Table 3 jcm-09-00056-t003:** Pretherapeutic laboratory values of patients with diagnosed ICC.

Variables	Unit	*n*	Median	Range
AFP	ng/mL	7	3.9	1.1–172.0
CA 19-9	kU/L	25	118	3.5–5821
Albumin	g/dL	36	4.2	3.1–5.0
Bilirubin	mg/dL	46	0.5	0.2–2.3
GGT	U/L	45	186	19–1165
AST	U/L	46	45	22–110
ALT	U/L	43	27	8–246
INR		46	1.0	0.9–1.5
Creatinin	mg/dL	46	0.8	0.5–1.6
MELD-Score		46	6	5–13

ICC = intrahepatic cholangiocarcinoma, AFP = alpha Fetoprotein, AST = Aspartate Transaminase, ALT = Alanin Transaminase, GGT = Gamma-Glutamyl-Transferase, INR= International Normalized Ratio, MELD= Model of End Stage Liver Disease, *n* = Numbers of patients where a value could be received (*n* = 46 in total).

**Table 4 jcm-09-00056-t004:** Procedural characteristics of TARE (transarterial radioembolization) therapy.

Variables	No. of Patients (Percentage %)
Site of SIR sphere application	
Monolobar	16 (34.8)
Bilobar selective	25 (54.3)
Bilobar superselective	3 (6.5)
Central position	2 (4.3)
Treated liver lobe	
Right liver lobe	14 (30.5)
Left liver lobe	3 (6.5)
Bilobar	29 (63.0)
Dose in GBq (median and range)	
Applied	1.74 (0.51–3.26)

**Table 5 jcm-09-00056-t005:** Results of follow-up at 3 and 12 months.

Variables	No. of Patients (Percentage %)	
	Follow-Up (3 months)	Follow-Up (12 months)
Status		
Alive	35 (76.1)	10 (21.7)
Dead	2 (4.3)	18 (39.1)
LTFU	9 (19.6)	18 (39.1)
Tumor response (RECIST)		
CR	0 (0.0)	0 (0.0)
PR	16 (34.8)	0 (0.0)
SD	7 (15.2)	5 (10.9)
PD	12 (26.1)	5 (10.9)

LTFU = Lost to follow-up, CR = complete remission, PR = partial remission, SD = stable disease, PD = progressive disease.

**Table 6 jcm-09-00056-t006:** Overall survival after TARE in dependence of pretherapeutic variables.

Variables	Number of Patients	Median Survival Time in Months	95% KI of OS in Months *	*p*-Value **
Sex				
Male	15	5.1	3.1–7.1	0.311
Female	22	11.2	7.9–14.5	
Systemic therapies before TARE				
Yes	24	7.3	3.1–11.6	0.054
No	13	18.8	E	
Resection				
Yes	6	3.9	2.4–5.3	0.002
No	31	12.8	5.1–20.6	
Liver cirrhosis				
Yes	5	***	***	0.384
No	32	8.7	5.4–11.9	
Tumorburden in the liver				
<25%	12	11.2	7.1–15.2	0.241
>25%	25	8.7	5.1–12.3	
Liver lobes				
Unilobar	14	15.6	9.0–22.2	0.042
Bilobar	23	6.4	3.5–9.3	
Portal Vein thrombosis				
Yes	30	7.3	3.1–11.6	0.796
No	7	9.5	2.1–17.7	
Extrahepatic Tumor				
Yes	12	7.3	4.6–10.1	0.295
No	15	9.5	6.0–13.0	
Hepatopulmonal shunting				
<10%	31	7.3	2.3–12.5	0.129
>10%	6	E	E	
Bilirubin				
Normal	35	9.5	6.4–12.6	0.715
Increased	2	5.1	E	

* Kaplan–Meier analysis, ** log-rank-test, *** analysis not performed as more than 50% of patients died within the observation period, E: analysis not performed due to limited cohort size.

**Table 7 jcm-09-00056-t007:** Independent Variables with a significant influence on overall survival.

Variables	Number of Patients	Hazard Ratio	95% CI for Hazard Ratio	*p*-Value *
Resection				
Yes	6	6.2	2.1–18.3	0.001
No	31	1	reference	reference
Liver lobe				
Unilobar	14	1	reference	reference
Bilobar	23	3.4	1.3–8.9	0.012

* Cox-Regression (forward and backward selection).

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
