# Peer review of "Prognostic Factors for Overall Survival in Advanced Intrahepatic Cholangiocarcinoma Treated with Yttrium-90 Radioembolization"

_jcm, 2019, doi:10.3390/jcm9010056_

Round 1

Reviewer 1 Report

The authors present a new series of patients treated with SIRT for ICC.

They combined 46 cases from 3 institutions, retrospectively analyzed, trying to define some prognostic parameters.

Some limitations of the present work limit the interpretation of the data.

Major comments:

1- Similar works have already been published, with series with higher numbers, and I don't feel the authors add significant novelties in the field justifying publication in a journal with large readership. The study would probably be more suitable for a speciality journal.

2- The main objective is presented to be the definition of prognostic factors. I'm not sure the work of the authors could actually answer this question. They study 46 patients, but only 37 are included in the prognostic model (lost to follow-up mainly, but 1 patient without explanation). The population is too limited to correctly define prognostic factors. For example, the use of previous stystemic treatment just missed statistical significance (p=0.054), but was reported by other series as an important prognostic factors. This is highly probable that lack of power is the cause of this discrepancy, and the conclusions of the authors might thus be misleading.

3- The authors included 3 patients with previous SIRT treatment. They could not be analyzed with the others, or the date of start of their observation should be the date of first SIRT. This is biaising their survival data.

4- Some clarification should be made in their methods. What does "bilobar selective" or "hyperselective" mean?

Minor comments:

1- the prognostic of ICC might be better than the figures given in the introduction or discussion. Please see Lamarca et al JNCI 2019.

2- A prospective trial was very recently published (certainly after submission of this manuscript), but should now be discussed: Edeline et al, JAMA Oncol 2019. Prognostic factors identified were lab values missing in the present manuscript.

Reviewer 2 Report

This study evaluating the safety, outcome and prognostic factors of radioembolization in intrahepatic cholangiocarcinoma is interesting since it includes a majority of patients with previous systemic therapy and radioembolization had not been tested in this setting. 

Major comments:

1) The title of the study is not concordant with the described aim of the study. The described aim (in the introduction and discussion) is to evaluate the safety and outcome in more advanced stages of the disease. However, the results of survival in patients with previous systemic therapy are not underlined in the discussion nor in the abstract, and are neither represented with a figure. 

2) Concerning the prognostic factors identified, it is surprising that the previous surgical resection of intrahepatic cholangiocarcinoma is associated with a significantly decreased survival. Indeed, Neuzillet et al. established on 405 patients a prognostic score of survival with second-line therapy in biliary tract cancers, where previous surgical resection was associated with a better survival (European Journal of Cancer 2019). This should be discussed and the authors should try to find explanations for these results. 

3) Authors should also study other prognostic factors such as: PS-ECOG score, other laboratory values than bilirubin and they should detail the nature of the extrahepatic tumor. One could argue that peritoneal carcinomatosis is of worse prognostic than lung or lymph nodes metastases.

4) The effective is little, with a lot of loss of follow-up (17%). We can suspect a lack of statistical power in order to show a difference in overall survival for patients with and without previous systemic therapy. The authors discussed this limitation.

Round 2

Reviewer 1 Report

The authors presented a revision of the previous version of their manuscript. They tried to address at the best of their possibilities the quality of the work.

However, major limitations remain:

Major comments:

1- The number are still quite low and lower than many previous publication in the field.

2- Due to this limitation, the claim to describe adequate prognostic factors could not be sustained.

Minor comments:

1- Please clearly state in the abstract the number of patients on which the main results are based, meaning the number of patients included in the MV model: 37.

2- Please add uncertainties in the conclusion of the abstract given the limitation of this manuscript.

Author Response

Major comments:

The number are still quite low and lower than many previous publication in the field.

RE: we answer to both point 1 and 2 below as both are closely related.

Due to this limitation, the claim to describe adequate prognostic factors could not be sustained.

RE: we absolutely agree with the reviewer that the number of cases in included, although pooled data from three centers are provided, are still low. With a total of n=46 patients a profound analysis of sub-cohorts and specific factors which might influence survival is definitely limited. An adequate description of prognostic factors, as correctly identified by the reviewer therefore has limited power. However, as not prospective register on radioembolization in ICC exist, and as the only prospective study (Edeline et al JAMA Oncol) similarly was restricted to n=41 patients, we still feel that our cohort presented and might contribute to subsequent meta-analyses.

In order not to over-interpret the meanings from the study with limited numbers we have modified the statements on prognostic factors especially in the results and discussion section of the manuscript in order to address the point made by the reviewer.

Minor comments:

Please clearly state in the abstract the number of patients on which the main results are based, meaning the number of patients included in the MV model: 37.

RE: as requested, we have added the actual number for the survival analyses (n=37) to the abstract.

Please add uncertainties in the conclusion of the abstract given the limitation of this manuscript.

RE: we agree with the reviewer that the conclusion of the abstract might be falsely interpreted, and we have modified that statement accordingly.

Reviewer 2 Report

The authors answered to all the questions that were raised and added the missing points to the manuscript. Make sure to perform an english minor spell check.

Author Response

The authors answered to all the questions that were raised and added the missing points to the manuscript. Make sure to perform an english minor spell check.

RE: we thank the reviewer for the positive response. Again, a spell check has been performed accordingly.